# Stromal Characteristics and Impact on New Therapies for Metastatic Triple-Negative Breast Cancer

**DOI:** 10.3390/cancers14051238

**Published:** 2022-02-27

**Authors:** Shelby A. Fertal, Johanna E. Poterala, Suzanne M. Ponik, Kari B. Wisinski

**Affiliations:** 1University of Wisconsin (UW) Carbone Cancer Center, Madison, WI 53705, USA; sfertal@wisc.edu (S.A.F.); jpoterala@uwhealth.org (J.E.P.); ponik@wisc.edu (S.M.P.); 2Department of Cell and Regenerative Biology, UW School of Medicine and Public Health, Madison, WI 53705, USA; 3Department of Medicine, UW School of Medicine and Public Health, Madison, WI 53726, USA

**Keywords:** triple-negative breast cancer (TNBC), immunotherapy, antibody–drug conjugates (ADCs), tumor microenvironment, stroma, extracellular matrix

## Abstract

**Simple Summary:**

In this review, we summarize the recent outcomes from clinical trials with new agents for metastatic triple-negative breast cancer, specifically focusing on immunotherapies targeting the PD-1/PD-L1 pathway and antibody–drug conjugates. In addition to the clinical data supporting these therapies, we review the impact of the tumor microenvironment on the effectiveness of these therapies.

**Abstract:**

The heterogenous nature of triple-negative breast cancer (TNBC) is an underlying factor in therapy resistance, metastasis, and overall poor patient outcome. The lack of hormone and growth factor receptors lends to the use of chemotherapy as the first-line treatment for TNBC. However, the failure of chemotherapy demonstrates the need to develop novel immunotherapies, antibody–drug conjugates (ADCs), and other tumor- and stromal-targeted therapeutics for TNBC patients. The potential for stromal-targeted therapy is driven by studies indicating that the interactions between tumor cells and the stromal extracellular matrix (ECM) activate mechanisms of therapy resistance. Here, we will review recent outcomes from clinical trials targeting metastatic TNBC with immunotherapies aimed at programed death ligand–receptor interactions, and ADCs specifically linked to trophoblast cell surface antigen 2 (Trop-2). We will discuss how biophysical and biochemical cues from the ECM regulate the pathophysiology of tumor and stromal cells toward a pro-tumor immune environment, therapy resistance, and poor TNBC patient outcome. Moreover, we will highlight how ECM-mediated resistance is motivating the development of new stromal-targeted therapeutics with potential to improve therapy for this disease.

## 1. Introduction

Triple-negative breast cancer (TNBC) makes up approximately 15–20% of all breast cancers [1]. TNBC refers to breast cancer that is lacking both hormone receptors, the estrogen receptor (ER) and progesterone receptor (PR), in addition to lack of Human Epidermal Growth Factor Receptor-2 (HER2) amplification. Due to the lack of these receptors, endocrine therapy or HER2-directed therapy is ineffective as a targeted therapy. This has led to chemotherapy being the mainstay of treatment in early-stage and metastatic TNBC. Chemotherapy is now usually recommended as preoperative (neoadjuvant) therapy with the goal of reducing the risk of recurrence and death. A subset of patients obtain a pathologic complete response (pCR) to neoadjuvant chemotherapy at the time of surgery, which is associated with an overall good prognosis and similar to that of other subtypes of breast cancer [2]. Unfortunately, those patients whose tumors do not achieve pCR have demonstrated poor event-free (EFS) and overall survival (OS) rates of 50–62% at 5 years [3,4]. Unlike hormone receptor-positive (HR+) disease, which can recur even decades following treatment, TNBC tends to recur within the first five years of initial treatment. After metastasis has occurred, survival is poor and averages approximately 13 months with chemotherapy [5,6].

It is noteworthy that TNBC is a very heterogeneous group of diseases. Using genomic profiling, Lehmann et al. initially identified six subtypes of TNBC [7]. After further analyses, there is now evidence that there are four distinct subtypes of TNBC [8]: luminal androgen receptor, mesenchymal, basal-like immunosuppressed, and basal-like immune-activated. These subtypes have unique prognoses, response to chemotherapy, and molecular targets. Despite these advances in understanding of TNBC heterogeneity, whether different treatment approaches should be recommended based on the subtype remains under investigation and subtype has not yet been incorporated into current TNBC treatment guidelines. 

Much of the heterogeneity of TNBC can also be attributed to the composition of the tumor microenvironment (TME), also known as stroma. The TME directly surrounds and continuously interacts with the tumor. It is comprised of vasculature, extracellular matrix (ECM), connective tissue, stromal fibroblasts, cytokines, and infiltrating immune cells. While specific components of the TME have been shown to play a role in disease progression [9,10,11], the development of cancer therapies often do not consider tumor interactions with the TME. Continued investigation into the interplay between components of TME and tumor cells is necessary to not only better understand the biology of metastatic TNBC but also to aid in the development of novel and effective therapeutics.

A major component of the TME is the ECM, a highly dynamic network of various macromolecules that form the structural environment in normal and diseased states that contributes to the mechanical properties of the tissue [12]. The main ECM molecules present in solid tumors, such as breast cancer, include fibular collagen, fibronectin, elastin, and laminins [13,14]. We and others have shown that increased ECM deposition in breast and other solid tumors is prognostic of poor patient outcome [15,16,17,18,19,20,21]. Poor outcomes are not only associated with changes in ECM composition. It is well appreciated that the alterations in collagen fiber architecture that accompany tumor progression (tumor-associated collagen signatures, TACS) are prognostic of poor patient outcome across all subtypes of breast cancer, including TNBC [16,20,22]. These alterations in fibrotic ECM direct cell behavior toward tumor progression [9,23,24], result in the exclusion of immune cell infiltration, promote immunosuppression and resistance to immune checkpoint inhibitor (ICI) therapy [25,26,27], and contribute to the limited distribution of chemotherapeutics that can cause clinical resistance [28].

The poorer prognosis of early-stage and metastatic TNBC is characterized by strong resistance to chemotherapy and radiotherapy; and although newer treatments are emerging for this disease, resistance continues to be a significant challenge. Many of the mechanisms of therapy resistance are driven by tumor cell interactions with the underlying composition and structure of the ECM. Specifically, the biophysical and biochemical cues from fibrotic ECM regulate the pathophysiology of tumor cell functions that are critical to metastatic progression, including invasion [29,30,31,32], epithelial to mesenchymal transition (EMT) [33,34,35], increased circulating tumor cells [36,37], and proliferation at the metastatic niche [26,38,39,40]. The fibrotic ECM also activates mechano-signaling in carcinoma-associated fibroblasts (CAFs) [41,42,43], leading to pro-tumor immune infiltration [17,20], and immunosuppressive cytokine signaling [20,26]. In addition, dense ECM creates a physical barrier that excludes T-lymphocytic cells, drives hypoxia, and reduces drug availability within the TME, all of which diminish the efficacy of cancer therapies [26,44,45]. The fibrotic ECM and immunosuppressive cues that drive breast cancer metastasis and therapy resistance have not only been identified in primary breast tumors, but they also occur at distal sights to regulate metastatic growth [26]. Based on this understanding of the ECM, there is mounting interest in developing therapies that target the stromal matrix [46,47,48]. 

In this review, we will examine recently approved new approaches to therapeutically target TNBC, focusing on metastatic breast cancer and the classes of immune checkpoint inhibitors (ICIs) and antibody–drug conjugants (ADCs). We will discuss the challenges that the TME poses to clinical therapies, specifically how the stroma drives immunosuppression and limits drug availability. Finally, we will highlight promising novel approaches to target the stroma in combination with immune modulation therapy or as direct molecular targets of ADC therapies.

## 2. Immunotherapy

While chemotherapy has historically been the backbone of treatment for TNBC [49,50,51], there has been recent interest in investigating the use of immunotherapy. This stems from several unique qualities of TNBC that make immunotherapy an effective target. TNBC has been found to have higher amounts of tumor-infiltrating lymphocytes (TILs) when compared to other subtypes of breast cancer [52]. TILs are immune cells that can be found in the tumor stroma and can have immune activity against the tumor cells. Higher stromal TIL content has been shown to predict improved clinical outcomes in patients with TNBC, such as longer progression-free survival (PFS) and OS in patients treated with neoadjuvant or adjuvant chemotherapy [52,53]. A higher level of TILs correlates to better responses seen with ICIs [54]. TNBC also has a higher level of expression of programmed death-ligand 1 (PD-L1) on tumor and immune cells [55,56] when compared to other breast cancer subtypes. PD-1 is an inhibitory immune checkpoint receptor expressed by activated T cells. When it binds to PD-L1 (B7-H1) or PD-L2 (B7-DC), this leads to immunosuppression by regulating the activity of effector T cells. Tumor expression of PD-L1 is one way that tumor cells evade the host’s immune system [57,58]. Inhibiting this ligand–receptor interaction via ICIs can lead to upregulation of the immune system and activation against tumor cells. Higher levels of PD-L1 expression are thought to correlate to a greater response from ICI [59]. For these reasons, immunotherapy has been studied in TNBC and has now emerged as a standard therapy. 

Results from early trials indicated that the safety profiles of the ICIs, atezolizumab [60] and pembrolizumab [61], were manageable but that there was limited benefit from single-agent therapy despite high PD-L1 expression or TILs. These findings led to further investigations in combination with chemotherapy. The IMpassion130 trial [62] (Table 1) studied the drug, atezolizumab (Tecentriq^®^, Genentech, South San Francisco, CA, USA) a monoclonal antibody which selectively targets PD-L1 and prevents the interaction with PD-1, thus inhibiting T-cell suppression. This was an international, phase III, randomized, double-blind, placebo-controlled trial comparing first-line atezolizumab plus nab-paclitaxel to placebo plus nab-paclitaxel in patients with locally advanced or metastatic TNBC. Nab-paclitaxel was used because, at the time of trial design, it was hypothesized that the glucocorticoid premedication given with paclitaxel may affect immunotherapy activity [63]. The two primary endpoints were PFS and OS. Median PFS in the intention-to-treat (ITT) population was found to be significantly longer in the atezolizumab-nab-paclitaxel group (7.2 months vs. 5.5 months; hazard ratio (HR), 0.80; 95% confidence interval (CI) 0.69 to 0.92; *p* = 0.002). The prespecified statistical analysis by PD-L1 status (PD-L1-positive ≥ 1% PD-L1 expression on tumor-infiltrating immune cells) also showed improved median PFS with atezolizumab (7.5 months vs. 5.0 months, HR 0.62; 95% CI 0.49 to 0.78; *p* < 0.001) [64]. Median OS in the ITT population was 21.0 months vs. 18.7 months (HR 0.87; 95% CI 0.75–1.02; *p* = 0.077) in the atezolizumab-nab-paclitaxel group and placebo plus nab-paclitaxel group, respectively. Due to the prespecified statistical plan, subsequent OS analysis in the PD-L1-positive population was exploratory but showed a median OS of 25.4 months vs. 17.9 months (HR 0.67; 95% CI 0.53–0.86) in the atezolizumab-nab-paclitaxel group and placebo plus nab-paclitaxel group, respectively. Importantly, the Kaplan–Meier curves, especially for the PD-L1-positive population, indicated an intriguing flattening of the curve around the 36 month mark, potentially signaling a durable response from treatment. While the incidence of grade 3 or 4 adverse events of special interest was higher in the atezolizumab-nab-paclitaxel group (7.5% vs. 4.3%), safety profiles were similar between the groups and side effects were largely manageable and did not lead to treatment withdrawal in the majority of patients. 

Following the IMpassion130 trial, IMpassion131 was completed [65] (Table 1), which tested atezolizumab and paclitaxel versus placebo and paclitaxel in patients with unresectable locally advanced or metastatic TNBC. This study found no improvement in investigator-assessed median PFS in the PD-L1-positive subgroup showing median PFS of 6.0 months in the atezolizumab-paclitaxel group vs. 5.7 months in the placebo-paclitaxel group. Final OS data also showed no difference between the arms (HR 1.11; 95% CI 0.76–1.64), showing a median OS of 22.1 months in the atezolizumab-paclitaxel group and 28.2 months in the placebo-paclitaxel group. When the ITT population was analyzed, similar results were found. 

Pembrolizumab (Keytruda^®^, Merck, Kenilworth, NJ, USA), a monoclonal antibody against PD-1, has also been studied in TNBC. The KEYNOTE-355 phase III randomized, double-blind trial [66] (Table 1) compared the efficacy and safety of pembrolizumab plus chemotherapy with placebo plus chemotherapy in patients with previously untreated unresectable locally recurrent or metastatic TNBC. Chemotherapy regimens consisted of nab-paclitaxel, paclitaxel, or gemcitabine plus carboplatin. The primary endpoints were PFS and OS in patients with combined positive score (CPS) of ≥10 and CPS of ≥1 and in the ITT population. CPS is defined as the number of PD-L1-positive cells (tumor cells, lymphocytes, and macrophages) divided by the total number of tumor cells × 100 [67]. In patients with tumors having a CPS ≥ 10, pembrolizumab improved median PFS to 9.7 months, compared to 5.6 months in the placebo-chemotherapy group (HR 0.65; 95% CI 0.49–0.86). In the ITT population, the median PFS in the pembrolizumab-chemotherapy group was also statistically higher at 7.5 months versus 5.6 months in the placebo-chemotherapy group (HR 0.82; 95% CI 0.69–0.97), although significance was not tested. The benefit on PFS in the pembrolizumab-chemotherapy group was noted across all pre-defined subgroups, regardless of choice of chemotherapy. There were more immune-mediated adverse effects in the pembrolizumab-chemotherapy group compared to placebo-chemotherapy (26% vs. 6%), but only grade 3 or higher in 5% of the patients in the pembrolizumab-chemotherapy group. Updated final analysis in 2021 showed an improvement in OS in the CPS ≥ 10 in the pembrolizumab-chemotherapy group (23.0 months vs. 16.1 months, HR 0.73; 95% CI 0.55–0.95; *p* = 0.009) [68]. This study’s inclusion of taxanes and a non-taxane platinum-based regimen allowed for a broader applicability and wider range of clinical scenarios where immunotherapy may provide clinical benefit. 

These are the largest trials to date assessing the efficacy of immunotherapy in advanced TNBC. In March 2019, the FDA provided accelerated approval of atezolizumab for its use in PD-L1-positive TNBC. This approval was contingent upon the results from the IMpassion131 trial, which failed to meet its primary endpoint of PFS superiority in this population. Therefore, in August 2021, the accelerated approval was withdrawn. In November 2020, the FDA approved pembrolizumab to be used in combination with chemotherapy for patients with PD-L1 CPS ≥ 10 positive unresectable or metastatic TNBC based off the results of the KEYNOTE-355 trial. 

It is worth noting that the trials used different PD-L1 assays. The IMpassion130 and 131 studies defined positive by tumor-infiltrating immune cells staining ≥ 1% utilizing VENTANA PD-L1 SP142 immunohistochemical testing [69]. With this assay, approximately 40% of metastatic TNBC are considered PD-L1 positive. The KEYNOTE-355 study used the PD-L1 IHC 22C3 pharmDx immunohistochemistry assay, and then characterized the samples by CPS, which includes tumor and immune cells. Using the cutoff of ≥10, 31–34% of newly metastatic TNBC and 60–65% of recurrent metastatic TNBC were found to be positive. While there is high concordance (80%) in patients screening positive by immune cell 1% and above via SP142 assay and via CPS ≥ 10, the assays should not be considered interchangeable [70]. In addition, intratumoral heterogeneity of PD-L1 expression and discordance between primary and metastatic sites remain key challenges with PD-L1 testing in TNBC [71,72].

The role of checkpoint inhibitors has also been an area of very active research in the non-metastatic TNBC setting. KEYNOTE-522 [73] was a phase III trial comparing neoadjuvant pembrolizumab plus chemotherapy versus placebo plus chemotherapy, followed by adjuvant pembrolizumab versus placebo. Initial results showed improved pCR with pembrolizumab and longer-term outcomes have now shown statistically significant improvement in event-free survival (EFS) with 3 year EFS 84.5% vs. 76.8% (HR 0.63, 95% CI, 0.48–0.82) [74,75]. In contrast to the metastatic setting, PD-L1 expression did not associate with benefit. This led to FDA approval of pembrolizumab for early-stage TNBC, regardless of PD-L1 status. The GeparNuevo study [76] was a phase II trial investigating neoadjuvant durvalumab with chemotherapy in early TNBC. This study showed an improvement in 3 year invasive disease-free survival (DFS) (HR 0.48; 95% CI 0.24–0.97), distant DFS (HR 0.31; 95% CI 0.13–0.74) and OS (HR 0.24; 95% CI 0.08–0.72). In contrast, the NeoTRIPaPDL1 study with atezolizumb versus placebo plus carboplatin and nab-paclitaxel chemotherapy did not meet its primary endpoint of improvement in pCR, so not all ICI studies have demonstrated success in non-metastatic disease [77]. Further research is needed to determine better biomarkers to determine which TNBC are most responsive to PD-1/PD-L1 inhibitors and whether novel combinations can reduce the chemotherapy component of the current regimens.

## 3. Tumor Microenvironment and Immune Modulation

While ICI for TNBC holds promise compared to other breast cancer subtypes, it is clear from clinical trials [62,65,66] that a better understanding of the mechanisms regulating ICI therapy response is required. Several studies have demonstrated that the heterogeneity in TIL infiltration and function, which is associated with ICI therapy response [54], is highly dependent on physical and chemical signals in the TME [78,79]. The aberrant accumulation and remodeling of collagenous ECM found in many solid tumors, including TNBC, is recognized as a crucial factor regulating the tumor immune response [48]. Highly dense ECM increases tumor stiffness. The tumor cells respond to stiffness by activating mechano-signaling pathways that lead to increased expression of immune regulatory factors, such as PD-L1 [80]. CAFs also respond to the mechanical stiffness of the TME by activating feedforward mechanisms to further enhance ECM deposition [41], resulting in poor diffusion, increased hypoxia, and metabolic stress [81]. All of these factors lead to the upregulation of immunosuppressive signaling molecules such as interleukin-10 (IL-10), chemokine ligand-2 (CCL2), CCL18, transforming growth factor β (TGFβ), and prostaglandin E2 [82,83,84,85,86]. The stiff, immunosuppressive TME not only impacts the behavior of breast cancer cells but also contributes to the recruitment and polarization of metastasis-promoting stromal cells such as tumor-associated macrophages (TAMs), and regulatory T cells (T-regs) to the primary tumor site [84]. Together, these studies demonstrate the importance of the TME in immune modulation, which has led to multiple approaches to target the stroma in combination with immune modulation therapy to improve patient outcome (Figure 1).

Attempts have been made to “normalize” the ECM and reduce tumor stiffness. Early approaches showed promise in pre-clinical studies but were hampered by off target effects and lack of specificity in clinical trials. Despite these setbacks, advances in pharmaceutical methods and insights into therapeutic timing have brought renewed interest in this approach. One key example is the lysyl oxidase (LOX) inhibitor, beta-aminopropionitrile (BAPN), which has been highly effective at reducing matrix stiffness in animal models. The transition of BAPN to clinical trial for breast cancer resulted in severe toxicity, precluding use in humans [87]. Taking a slightly different approach, Takai et al. observed that inhibition of TGFβ, a signaling molecule that activates CAFs to secrete collagen, leading to tumor fibrosis, may decrease tumor growth and metastasis. When Pirfenidone, a TGFβ antagonist and antifibrotic agent, was administered in conjunction with doxorubicin, a standard of care chemotherapeutic, there was a statistically significant decrease in tumor growth and lung metastasis [88]. However, when used in combination with immunotherapy in pre-clinical breast cancer models, inhibition of TGFβ attenuated immune modulation therapy [89]. Thus, the need for alternate approaches to therapeutically target the ECM remains. 

One such alternate approach demonstrated that a common angiotensin II type 1 receptor blocker, losartan, increases tumor perfusion and decreases hypoxia by reducing the amount of ECM in the TME (Figure 1B). This reduction in ECM is due to the inhibition of TGFβ signaling, downstream of angiotensin II type 1 receptors, when losartan is bound [90]. The combination of losartan and radiotherapy significantly decreased lung metastasis and increased host survival by increasing the number of functional tumor vessels and reducing TME hypoxia [91]. In TNBC patients, there is a current phase II clinical trial testing the safety and effectiveness of camrelizumab, a PD-1 inhibitor, in combination with liposomal doxorubicin and losartan in patients that have not received more than one prior line of chemotherapy and are advanced or locally advanced (NCT05097248) (Table 2) (Figure 1B).

Aside from directly inhibiting ECM deposition, there is interest in targeting stomal cell populations such as tumor-associated macrophages (TAMs). While the level of TILs is associated with a better prognosis for patients, high levels of infiltrating TAMs in TNBC are indicative of a poorer prognosis [92]. The infiltration of pro-tumor TAMs results in an increase in angiogenesis, matrix remodeling, immunosuppression and tumor cell invasion [93]. The stiff regions of ECM localized at the invasive front of breast tumors, which are associated with aggressive disease, are enriched in aligned collagen fibers and CD163+ TAMs [17,94]. In pre-clinical models of collagen-dense mammary carcinoma, increased levels of the inflammatory mediators, CCL2 and cyclooxygenase-2, were accompanied by an increase in TAM infiltration [85]. CCL2 recruits Tie2-receptor-expressing macrophages that facilitate the trafficking of tumor cell along aligned collagen fibers toward the endothelium where tumor cell dissemination occurs [95,96,97]. Thus, Tie2-receptor-positive TAMs have become an attractive target for therapy. The development and pre-clinical testing of a specific Tie2-receptor inhibitor, rebastinib, demonstrated high therapeutic potential by blocking recruitment of Tie2+ macrophages, and reducing circulating tumor cells and metastatic lesions (Figure 1A) [98]. Currently, rebastinib is in a phase I clinical trial in combination with microtubule-targeting agents for patients with metastatic breast cancer (NCT02824575) (Table 2).

Within the tumor microenvironment, CAFs synergize with TAMs to accelerate cancer progression, resulting in poor prognosis [99]. CAFs are the most common non-epithelial cell type in the TME and another potential target for new therapies. Breast cancer cells recruit and activate CAFs through the secretion of growth factors such as TGFβ and fibroblast growth factor-2 [24]. CAFs can also be activated directly by mechanical cues from the ECM. CAFs are the primary cell type involved in matrix deposition and remodeling [43,100,101,102]. CAFs also communicate with both tumor cells and immune cells through secreted factors to regulate immune suppression [103]. For example, CAFs secrete CXCL12 to attract and retain CD4+CD25+ T cells and induce differentiation of these cells into T-regs. The increased number of T-regs inhibits the proliferation and lytic function of CD8+ T-lymphocytes [104,105]. The expression of CAF-specific cell surface markers, such as fibroblast activation protein (FAP), has opened the door for attempts to target this stromal cell population to reduce immunosuppressive signaling. Currently, antibodies against FAP have been developed to selectively target CAFs and there are several phase I clinical trials underway that utilize this approach for breast cancer therapy. RO6874281 is a novel, monomeric, bispecific IL-2v immunocytokine that binds FAP on CAFs with high affinity. The IL-2v domain activates IL-2 receptor expressed on CD8+ T cells and natural killer (NK) cells independent of FAP binding. Importantly, targeting FAP retains IL-2v within the tumor microenvironment, which selectively promotes an anti-tumor immune response. Currently, RO6874281 is in a phase I dose escalation trial as either a single-agent or in combination with trastuzumab or cetuximab (NCT02627274) (Table 2).

Developing treatments directed toward cytokine signaling in the TME is another approach to block stomal cell function in breast cancer. Several immunosuppressive cytokines, including IL-10 and IL-4, are secreted from tumor cells and known to polarize TAMs to a pro-tumor phenotype and are associated with poor patient prognosis [92,106]. In contrast, IL-12 functions to promote many anti-tumor properties such as inhibiting angiogenesis, increasing the activation and survival of memory T cells, inducing adaptive immunity, and inhibiting T-helper 2 and T-regs [107]. The secretion of IL-12 results in a proinflammatory immune response and is currently the focus of a phase II Keynote-890 trial. This trial is being conducted in patients with inoperable advanced TNBC and it involves an intratumoral injection of tavokinogene telseplasmid, a plasmid encoding for IL-12, followed by electroporation and pembrolizumab (NCT03567720) (Table 2). The electroporation in the same region of the tumor as the tavokinogene telseplasmid injection results in the destabilization of the tumor cell membrane, increasing the uptake of IL-12 [107]. The proinflammatory tumor environment now is more favorable to being targeted by an anti-PD-L1 drug, such as pembrolizumab, and further demonstrates the importance of utilizing components of the TME to target cancer. 

Another clinical trial utilizing the pro-inflammatory nature of IL-12 to target the TME for improved drug response is underway. This phase I clinical trial targets patients with HR+, HER2-negative breast cancer with a treatment combination of NHS-IL12 and bintrafusp alfa in addition to radiation therapy. NHS-IL12 is a tumor-targeting immunocytokine that results in the delivery of IL-12 to the TME, while bintrafusp alfa is a bifunctional fusion protein that targets both TGFβ and PD-L1 by fusing the extracellular domain of TGFβ receptor II to a human immunoglobulin G1 antibody blocking PD-L1 (NCT04756505) (Table 2) [108,109]. Bintrafusp functions as a TGFβ “trap”, resulting in a combined reduction in TGFβ signaling within the TME and inhibition of the immune checkpoint with anti-PD-L1 action that reduced tumor growth to a greater extent than either element individually (Figure 1C) [108]. As depicted in Figure 1C, anti-TGFβ treatment reduces the fibrous nature of the ECM, allowing enhanced diffusion of anti-PD-L1 to block the signaling between T cells and tumor cells. The combination of NHS-IL12 and bintrafusp alfa has the advantage of IL-12-induced adaptive immune response and inhibition of the TGFβ signaling pathway to reduce tumor burden [109]. Bintrafusp alfa is also being tested alone in a separate phase I clinical trial that is recruiting patients with stage II–III HER2-positive breast cancer (NCT03620201) (Table 2) and in a phase II trial for TNBC (NCT04489940) (Table 2).

## 4. Antibody–Drug Conjugates

In addition to immunotherapy, current research for TNBC is also focusing on a new class of drugs classified as antibody–drug conjugates (ADCs). ADCs are composed of a monoclonal antibody linked to a cytotoxic drug, which is called “the payload”. The intent of ADC therapy is to allow for a more targeted chemotherapy delivery by the payload being brought specifically to the antigen-expressing cells and thus achieving a goal of higher anti-tumor efficacy and lower toxicity to non-malignant tissue. Here, we discuss the current landscape for this class of drugs with metastatic TNBC. 

Trophoblast cell-surface antigen 2 (Trop-2) is a transmembrane calcium signal transducer [110] and an epithelial cell adhesion molecule (EpCAM) family member [111] that has been detected in healthy epithelial cells of many organs. Recently, it has been identified to be upregulated in many tumor types, including TNBC [112]. There are two pools of Trop-2: one localized in the cell membrane and one in the cytoplasm. Interestingly, membrane-associated Trop-2 has been shown to be an unfavorable prognostic factor for OS, while the intracellular Trop-2 has a favorable impact on prognosis [113]. Sacituzumab govitecan (Trodelvy^®^ Gilead, Foster City, CA, USA) is an ADC comprised of an anti-Trop-2 IgG1 kappa antibody coupled to SN-38, which is a topoisomerase I inhibitor and an active metabolite of irinotecan [114]. These are connected via a hydrolysable linker [110,115]. The drug works by first binding to the Trop-2 expressed on the tumor cell membrane [110]. This allows for targeted delivery of SN-38 to the tumor cell after internalization of the ADC as well as to close by tumor cells via the bystander effect since SN-38 is also membrane permeable [112,116]. The SN-38 works by preventing ligation of cleaved DNA strands, leading to double-strand DNA breaks and ultimately, cell death [117]. 

A phase I dose-finding trial found that sacituzumab govitecan had acceptable toxicity and encouraging therapeutic activity in a variety of advanced solid cancers [118]. A phase I/II basket design trial, IMMU-132 [119], included patients with heavily pretreated (median of five lines of prior therapy) metastatic TNBC. The patients received 10 mg/kg dosing on days 1 and 8 of 21 day cycles. Overall, 30% achieved confirmed objective responses (partial response *n* = 19; complete response *n* = 2) with a median response duration of 8.9 months. A median time of 1.9 months to objective response was found. Median PFS was 6.0 months and OS was found to be 16.6 months. It was found that 41% of patients developed one or more grade 3 adverse events. This was mostly neutropenia (39%), but leukopenia (16%), anemia (14%), and diarrhea (13%) were also noted. This was followed by the IMMU-132-01 trial [120], which was an additional phase I/II single-group, multicenter trial. This included 108 patients with pretreated (median of three prior lines of therapy) metastatic TNBC. The same treatment regimen was given. This study showed a response rate of 33.3% and median duration of response of 7.7 months. Median PFS was 5.5 months and OS was 13.0 months. Adverse events were similar to those seen in the IMMU-132 trial with neutropenia and anemia being the most common. 

These early-phase trial results led to further investigation with the ASCENT trial [121] (Table 1). This was a phase III, international, open-label, randomized trial. It included patients with metastatic TNBC that were relapsed or refractory to ≥2 standard chemotherapy regimens. Previous therapy had to include a taxane. Patients with brain metastases were excluded from the primary endpoint analysis. Patients were randomly assigned in a 1:1 ratio to receive sacituzumab govitecan or single-agent chemotherapy, which was determined prior to randomization and included eribulin, vinoreline, capecitabine, or gemcitabine. Treatment was continued until disease progression, unacceptable toxic effects, withdrawal from trial, or death. The primary endpoint was PFS, as determined by blinded independent central review, among patients without known brain metastases, although screening for brain metastasis was not required. Secondary endpoints included OS, PFS by investigator assessment, objective response, and safety. A total of 468 patients were enrolled as the primary trial population (235 assigned to receive sacituzumab govitecan and 233 to receive single-agent chemotherapy). At the time of data cutoff, the median PFS, as determined by central review, was 5.6 months in the sacituzumab govitecan group and 1.7 months in the chemotherapy group (HR for disease progression or death 0.41; 95% CI 0.32–0.52; *p* < 0.001). This was consistent with the investigators’ assessment of PFS. The median OS was 12.1 months in the sacituzumab govitecan group and 6.7 months in the chemotherapy group (HR 0.48; 95% CI 0.38–0.59; *p* < 0.001). Subgroup analyses of median PFS and OS all favored sacituzumab govitecan over chemotherapy. An objective response rate of 35% was seen with sacituzumab govitecan compared to 5% with chemotherapy. The median duration of response was 6.3 months with sacituzumab govitecan and 3.6 months with chemotherapy. The most common adverse events were neutropenia (63% with sacituzumab govitecan and 43% with chemotherapy), diarrhea (59% with sacituzumab govitecan and 12% with chemotherapy), and nausea (57% with sacituzumab govitecan and 26% with chemotherapy). The most frequent treatment-related adverse event that was grade 3 was neutropenia (51% with sacituzumab govitecan and 33% with chemotherapy). Adverse events leading to discontinuation of therapy was 5% in both populations. 

The results seen in the IMMU-132 trial provided the basis for accelerated approval by the FDA in April 2020 for patients with unresectable locally advanced or metastatic TNBC who have received two or more prior systemic therapies pending the results of the confirmatory ASCENT trial. Once the results of the ASCENT trial were available, in April 2021, the FDA granted regular approval of sacituzumab govitecan for patients with unresecatable or metastatic TNBC who have received two or more prior lines of systemic therapy.

There are several ongoing early-phase trials assessing the safety and efficacy of sacituzumab govitecan in combination with other treatment approaches for metastatic TNBC. These include sacituzumab govitecan with rucaparib (NCT03992131) and talazoparib (NCT 04039230), both different PARP inhibitors, as well as sacituzumab govitecan with ICIs, pembrolizumab (NCT 04468061) and atezolizumab (NCT03424005). 

Although no other ADCs are approved for TNBC, several are under investigation. Glembatumumab-vedotin (GV) is an ADC consisting of a fully-human glycoprotein non-metastatic B (gpNMB)-specific IgG_2_ antibody couple with a microtubule inhibitor, monomethyl auristatin E (MMAR + E) via a protease-sensitive valine-citrulline peptide linker. GpNMB is a transmembrane protein overexpressed in approximately 40% TNBC and is associated with a poor prognosis [122]. After initial safety data were gathered, the EMERGE study looked at the activity of GV in heavily pretreated advanced breast cancer [123]. Analysis suggested that patients with advanced TNBC that overexpress gpNMB were most likely to derive benefit from GV. This led to the METRIC study, which was a phase IIb study including patients with advanced TNBC that overexpressed gpNMB (25% cells). The patients were randomized to receive GV (213 patients) or capecitabine (92 patients). Unfortunately, this study did not meet its primary endpoint with a median PFS of 2.9 months (95% CI 2.8–3.5) in the GV group and 2.8 months (95% CI 1.6–3.6; *p* = 0.76) with capecitabine. There were also no differences in the secondary outcomes of OS, overall response rate, or duration of response and therefore, GV is no longer being pursued. 

Trastuzumab-deruxtecan (T-DXd), which is made up of the well-established HER2-directed antibody (trastuzumab) and a topoisomerase I inhibitor conjugate, is approved for the treatment of HER2+ metastatic breast cancer, but emerging data have indicated activity in HER2low tumors, including TNBC. In a recent analysis of the DAISY study presented at the San Antonio Breast Cancer Symposium in December, 2021, T-DXd showed a 38% response rate in patients with low HER2 expression and an almost 30% response rate in patients without detectable expression of HER2 [124]. This agent is also being studied in the BEGONIA trial (NCT03742102), a phase Ib/II open-label multicenter study to determine the efficacy and safety of durvalumab in combination with novel oncology therapies. Trastuzumab-deruxtecan is added in one of the arms, which will consist of patients with advanced, unresectable, or metastatic TNBC that have HER2 low tumor expression. 

Additional ADCs that are currently being studied in early-phase trials for metastatic TNBC include ladiratuzumab-vedotin (humanized IgG_1_ antibody directed against LIV-1) with pembrolizumab (NCT03310957), U3-1302 (humanized anti-HER3 antibody linked with a topoisomerase I inhibitor, NCT04699630), CAB-ROR2-ADC (conditionally active biologic ROR2-targted ADC, NCT03504488), and anti-CA6-DM4 immunoconjugate (humanized DS6 antibody directed against tumor-associated sialoglycotope CA6 conjugated to maytansinoid DM4, NCT02984683). It is anticipated that ADCs will continue to change to landscape of treatment options for TNBC.

## 5. Tumor Microenvironment and ADCs

ADCs are one of the fastest-growing anti-cancer therapies. However, a primary hurdle of this therapeutic approach is effective and homogeneous penetrance of ADC delivery into solid tumors. It has been demonstrated that when an ADC is injected into the blood stream of humans, only a small fraction is delivered to the tumor [125]. Subsequently, only a fraction of the ADC that arrives in the TME actually binds to the tumor-specific cells [125,126]. The increase in ECM deposition contributes to the highly dense nature of breast tumors, which significantly impacts drug transport in both direct and indirect ways that ultimately lead to therapy resistance. Limited diffusion of ADCs into the TME is often observed in highly cross-linked ECM. The physical barrier function of the highly cross-linked ECM is most notably due to the family of five lysyl oxidase (LOX) isoenzymes which catalyzes the formation of cross-links within collagen and elastin molecules [127]. The enzyme activity of LOX is essential for the generation of insoluble fibers and the stabilization of collagen. Moreover, high expression of collagens I–V, LOX and LOXL1-2 in cancers, including breast carcinoma, is associated with therapy resistance [128]. LOX-induced crosslinking enhances the accumulation of collagen and increases the stiffness of the TME to further decrease diffusivity [25]. Treatment with LOX inhibitors in pre-clinical breast cancer models has been highly effective to increase drug penetration [128]. While LOX inhibition was unsuccessful in prior clinical trials [87], the concept of targeting matrix-modulating enzymes to improve drug penetration remains rich in therapeutic potential. 

Alteration in tumor vasculature is another cause of variability in drug distribution within solid breast tumors. While normal vessels within an organ are observed to organize in an orderly fashion, tumor vessels are better characterized as chaotic in their organization and the level of chaos continues to increase during disease progression [25]. This leads to an increase in heterogeneity of blood vessels within the tumor and, as a result, leads to various parts of the tumor receiving inconsistent blood flow. For example, one section of tumor might experience a rush of blood while perfusion to another region is more stagnant. The lack of perfusion to certain areas of a solid tumor also increases the incident of hypoxia. The variability in blood flow can be attributed to either solid stress, a physical force, or vessel leakiness. The underlying cause of all factors is an increase in the deposition of fibrous ECM in the TME [25,28]. Primeau et al. investigated how vascular density and blood flow influences the delivery of chemotherapy, specifically doxorubicin, in solid tumors. It was observed that the concentration of doxorubicin decreases exponentially with distance from tumor blood vessels [28]. This lack of diffusion of the drug was still observed even at high doses and viable cancer cells, cells outside of a hypoxic region, were not exposed to doxorubicin [28]. The limited drug penetration is also suggestive of the fact that few cancer cells will be sufficiently exposed to the chemotherapy to undergo apoptosis. These findings reveal the significance of an increase in ECM deposition to limit drug delivery, driving the need for anti-fibrotic agents to be administered in combination with ADC therapies. 

In recent years, tremendous effort has been put toward increasing the tumor penetrability of an ADC [45,129]. Due to their large molecular weight, many ADCs have difficulty diffusing from the vasculature, through the ECM, and then penetrating a solid tumor. With current practices, it is assumed that a drug distributes homogenously throughout the tumor site; but in practice, dosing is heterogeneous throughout not only a tumor site but metastatic lesions as well. This heterogeneity leads to the failure of a portion of neoplastic cells to receive the therapeutic dose of a drug and this can be exacerbated by the large molecular weights of ADCs [125,126] The larger molecular weights also impact how ADCs are eliminated by the body due their reliance on endocytosis and cellular interaction. Smaller molecules can be cleared via more traditional pathways such as hepatic excretion and renal elimination [130]. These smaller versions of ADCs utilize nanobodies to form nanobody–drug conjugates (NDCs), which can rapidly diffuse into the tumor. However, NDCs also utilize traditional clearance pathways, resulting in reduced bioavailability due to rapid clearance [131]. This can be mitigated by conjugating NDCs to carriers such as polyethylene glycol (PEG) or albumin. By conjugating to PEG, NDCs experience improved drug solubility, circulation, reduced immunogenicity, and controlled release [131,132].

Unlike traditional antibody-targeted therapeutics, which depend and assume on immediate internalization of the drug into the tumor cell where it can act on its intended target, ADCs targeting and binding surface receptors on stromal cells or the ECM directly have the potential to accumulate and form a chemotherapeutic depot. This system is dependent on determining a target within the stroma that is unique enough from normal stroma to allow for selectivity and reduce cytotoxic effects from nonspecific binding. As mentioned above, FAP is a unique CAF marker utilized as a stromal target. Currently, a FAP-activated doxorubicin pro-drug is being tested in a phase I open-label clinical trial for breast and other solid tumors (NCT04969835) (Table 2). Another possible target that has emerged in other cancer systems is tenascin-C. Tenascin-C is a highly expressed protein in breast cancer ECM, as well as ovarian and lung cancer, while showing extremely low expression in the ECM of healthy tissue. Dal Corso et al. utilized a F16-PNU159682 as a delivery system targeting the ECM protein, tenascin-C [133]. F16 is an antibody specific to the alternatively spliced A1 domain of tenascin-C, and is able to effectively deliver PNU159682, a novel anthracycline, specifically to the tumor stroma (Figure 1D) [133]. Treatment with F16-PNU159682 resulted in a halt in tumor growth for 20 days following injection into A431 epidermoid tumor-bearing BALB6 mice. These results are promising for non-internalized payloads and demonstrate the broad potential of ECM-targeted ADCs for breast cancer and other ECM-dense tumors.

## 6. Conclusions

Triple-negative breast cancer is a heterogeneous disease for which treatment options had traditionally been limited to chemotherapy. However, understanding this heterogeneity and the biology of this disease has resulted in new therapies which have improved the prognosis for patients impacted by TNBC. Immune checkpoint inhibitors and antibody–drug-conjugates are FDA approved and widely used in the treatment of metastasis but have had limited success. Components of the TME have emerged as a mediator of resistance to these novel therapies and as potential therapeutic targets for the future. Leveraging the growing knowledge on stromal–tumor interactions will hopefully continue to improve outcomes for patients with TNBC.

## Figures and Tables

**Figure 1 cancers-14-01238-f001:**
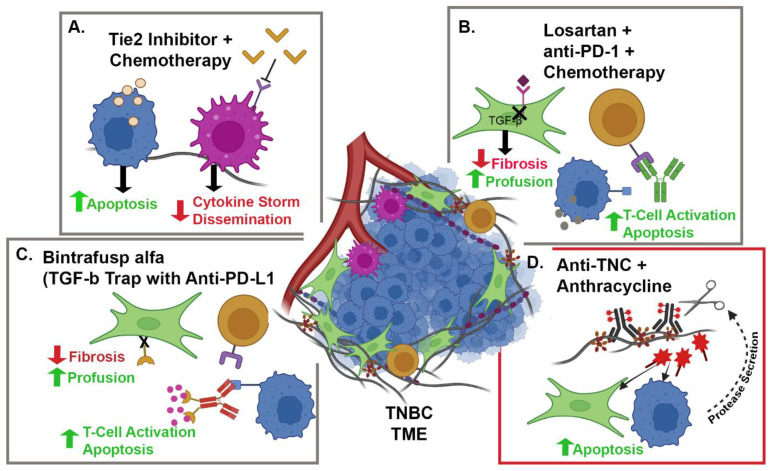
Schema of fibrotic TME and the stroma-targeted therapies currently undergoing clinical trials. (**A**) Tie2 inhibitors (tan arrow heads) block the Tie2 receptor site, resulting in a decrease in cytokine storm, dissemination, and inhibits macrophage (purple) tumor cell (blue) interactions along collagen fibers (gray). This results in a reduction in Tie2+ TAMs present in the TME. A chemotherapy (tan), such as paclitaxel, interacts with breast cancer cells (blue,) leading to an increase in apoptosis and a reduction in tumor burden. (**B**) Losartan (purple diamond) blocks angiotensin II binding to the angiotensin II type 1 receptor (purple) on cancer-associated fibroblasts (green), which inhibits downstream TGFβ signaling. This results in a decrease in fibrosis and a decrease in hypoxia due to an increase in vascularization. The reduction in fibrosis and increase in vascularization lend way to improved perfusion of drugs such as anti-PD-1 and chemotherapies. Anti-PD-1 immunotherapies (green antibody), such as camrelizumab, block the binding of PD-L1 (blue) to PD-1 (light purple), leading to activation of T cells (dark tan). Chemotherapies (gray), such as doxorubicin, act on the breast cancer cells (blue) to increase apoptosis. (**C**) The immunotherapy Bintrafusp alfa is a bifunctional protein that contains an antibody blocking PD-L/PD-L1 (red) interactions and the extracellular domain of TGFβ receptor II, resulting in a “TGFβ trap” (pink). Anti-PD-L1 results in an increase in T-cell activation and apoptosis of the cancer cell. The “TGFβ trap” reduces the concentration of extracellular TGFβ, resulting in a decrease in TGFβ signaling in cancer-associated fibroblasts (green), which causes a reduction in fibrosis. (**D**) An ADC (red box) against tenascin-C carrying an anthracycline, such as F16-PNU159682, binds the ECM protein tenascin-C (brown). Tumor-secreted proteases cleave the anthracycline (red stars) from the antibody, releasing the drug in the TME. This results in the endocytosis of anthracycline by TME cells, leading to in an increase in apoptosis of not only breast cancer cells (blue) but also stromal cells, such as cancer-associated fibroblasts (green). Created with BioRender.com.

**Table 1 cancers-14-01238-t001:** Summary of Recent Trials for Novel Therapeutics in Metastatic TNBC.

Study	Study Groups	Line of Therapy	Total Number of Patients	Study Design	Progression-Free Survival	Overall Survival	Response Rate
Immunotherapy							
IMPassion130 ^%^	Atezolizumab + nab-paclitaxel vs. placebo + nab-paclitaxel	1st	902	Phase III, randomized, double-blind, placebo-controlled trial	7.2 vs. 5.5 months (*p* = 0.002)	21.3 vs. 17.6 months (*p* = 0.08)	56.0% vs. 45.9%
IMPassion131 ^^^	Atezolizumab + paclitaxel vs. placebo + paclitaxel	1st	651	Phase III randomized, double-blind, placebo-controlled trial	5.7 vs. 5.6 months	19.2 vs. 22.8 months	
KEYNOTE-355 ^&^	Pembrolizumab + chemotherapy ^$^ vs. placebo vs. chemotherapy	1st	847	Phase III randomized, double-blind, placebo-controlled trial	9.7 vs. 5.6 * months	23.0 vs. 16.1 * months (*p* = 0.009)	52.7% *
Sacituzumab govitecan							
ASCENT ^+^	Sacituzumab govitecan vs. chemotherapy ^#^	≥2 prior	468	Phase III, Randomized	5.6 vs. 1.7 months (*p* < 0.001)	12.1 vs. 6.7 months (*p* < 0.001)	35% vs. 5%

* In CPS ≥ 10 group. All other reported statistics for intention-to-treat groups. ^$^ Carboplatin + gemcitabine, paclitaxel or nab-paclitaxel. ^#^ eribulin, vinorelbine, gemcitabine, or capecitabine. ^%^ https://clinicaltrials.gov/ct2/show/NCT02425891 (accessed on 18 January 2022); ^^^ https://clinicaltrials.gov/ct2/show/NCT03125902 (accessed on 18 January 2022); ^&^ https://clinicaltrials.gov/ct2/show/NCT02819518 (accessed on 18 January 2022); ^+^ https://clinicaltrials.gov/ct2/show/NCT02574455 (accessed on 18 January 2022).

**Table 2 cancers-14-01238-t002:** Summary of Recent Trials for Novel Therapeutics Targeting Stroma.

Clinical Trial Identifier	Study Groups	Cancer	Stromal Target	Study Design	Pre-Clinical Reference
Immunotherapy					
NCT05097248	Camrelizumab + Liposomal Doxorubicin + Losartan	TNBC	CAFs and PD-1	Phase II, single-arm, open-label, prospective clinical trial	88
NCT02824575	Paclitaxel + Rebastinib vs. Eribulin + Rebastinib	BC	TIE-2 Expressing Macrophages	Phase I non-randomized, open-label clinical trial	95
NCT03567720	Pembrolizumab + Tavo + EP vs. Pembrolizumab + Tavo + EP + Nab-Paclitaxel	TNBC	IL-12 and PD-L1	Phase II non-randomized, open-label, multicohort clinical trial	97
NCT04756505	Bintrafusp alfa + NHS-IL-12 + Radiation	HR+, HER2 − BC	IL-12, PD-L1 and TGFβ	Phase I, open-label clinical trial	98 and 99
NCT03620201	Bintrafusp alfa + chemotherapy	HER2+ BC	PD-L1 and TGFβ	Phase I, open-label clinical trial	98 and 99
NCT04489940	Bintrafusp alfa	TNBC	PD-L1 and TGFβ	Phase II, open-label clinical trial	98 and 99
Antibody–Drug Conjugates					
NCT04969835	AVA6000	BC and Solid Tumors	FAP	Phase I, open-label, 3 + 3 clinical trial	Avacta Life Science Ltd.

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
