# Peer review of "Stromal Characteristics and Impact on New Therapies for Metastatic Triple-Negative Breast Cancer"

_cancers, 2022, doi:10.3390/cancers14051238_

Round 1
Reviewer 1 Report
Reviewer Comments:
Shelby A. Fertal and coworkers present the manuscript entitled “Stromal characteristics and impact on new therapies for metastatic triple negative breast cancer”. The authors presented an interesting up to date which touches upon most of the relevant concepts and is of added value to the field. However, I have some minor concerns that need to be addressed before potential publication of the manuscript.
Minor comments:
On lines 52 to 55, several molecular subtypes were already abbreviated on the lines above, please make sure to use the abbreviations.
Based on the references reviewed, do you consider that there are 6 or 4 molecular subtypes of triple negative breast cancer?, its confusing to understand how many types of breast cancer subtypes exist.
Line 99. Authors mentioned two processes in the review's introduction: stromal regulation of tumor progression and metastasis. However, the issue of the ECM's effect on metastasis or the epithelial-to-mesenchymal transition receives minimal attention, and the involvement of extracellular Matrix Alterations in Metastatic Processes were not described throughout the article.
On line 113. Are there differences in PD-L1 expression in the different molecular subtypes of triple-negative breast cancer? A table showing the expression profiling data of PDL1 in several types of tumors, it will be useful and informative.
Line 230. In terms of matrix stiffness, it is important to mention how stiffness promotes the reorganization of tumor cell cytoskeleton and chromatin via cell-extracellular matrix and communication mechanisms such as SUN-KASH domain proteins to influence gene expression programs. This might explain, in part, the activation of immunosuppressive signaling molecules, in addition to the previously described variables (hypoxia, metabolic stress).
Line 273. It will be very useful and informative if authors briefly describe the potential mechanisms of action of losartan in tumor cells or how the renin-angiotensin system is connected to the impacts of tumor perfusion, hypoxia, or ECM in the tumor microenvironment.
On line 349, where they mention that "anti-TGFβ treatment reduces the fibrous nature of the ECM". It is not clear what is indicated in Figure 1. In addition, it is important to mention which part of Figure 1 they are referring to.
Table 1. Summary of Recent Trials for Novel Therapeutics in Metastatic TNBC. Please add the URL for the web page register in https://clinicaltrials.gov/, if any.
Finally, feel that the review is very complex and involves several factors, so it would be more clear if authors divide the Figure 1 into two parts: i) addressing all the mechanisms of the tumour microenvironment and immune modulation, ii) the mechanisms of the tumour microenvironment and ADCs.
Author Response
We thank the reviewers for their helpful comments, which have improved the clarity and overall flow of our manuscript. We have addressed the reviewers concerns in a point-by-point discussion below. The changes are also highlighted in red in the revised manuscript.
Reviewer #1:
Shelby A. Fertal and coworkers present the manuscript entitled “Stromal characteristics and impact on new therapies for metastatic triple negative breast cancer”. The authors presented an interesting up to date which touches upon most of the relevant concepts and is of added value to the field. However, I have some minor concerns that need to be addressed before potential publication of the manuscript.
Minor comments:
On lines 52 to 55, several molecular subtypes were already abbreviated on the lines above, please make sure to use the abbreviations.
---Abbreviations were removed as they are not mentioned again later in the manuscript.
Based on the references reviewed, do you consider that there are 6 or 4 molecular subtypes of triple negative breast cancer?, its confusing to understand how many types of breast cancer subtypes exist.
---Thank you for pointing out this lack of clarity. We have updated the manuscript which now reads, “Using genomic profiling, Lehmann et al. initially identified six subtypes of TNBC [7]. After further analyses, there is now evidence that there are four distinct subtypes of TNBC [8]: luminal androgen receptor, mesenchymal, basal-like immunosuppressed, and basal-like immune-activated. These subtypes have unique prognoses, response to chemotherapy, and molecular targets.”
Line 99. Authors mentioned two processes in the review's introduction: stromal regulation of tumor progression and metastasis. However, the issue of the ECM's effect on metastasis or the epithelial-to-mesenchymal transition receives minimal attention, and the involvement of extracellular Matrix Alterations in Metastatic Processes were not described throughout the article.
---We thank the reviewer for this comment. Our intent was to provide information on the pathophysiological changes in tumor cells in response to the fibrotic ECM to demonstrate the important of the ECM in breast cancer. However, our main point is to highlight that the underlying cues from the stroma drive immunosuppression and limit drug availability, and thus the stroma is rich in therapeutic potential. A discussion of the biophysical signaling mechanisms that regulate tumor progression and metastasis is outside the scope of this review. We have modified the text to clarify this point: Page 2-3, “The fibrotic ECM and immunosuppressive cues that drive breast cancer metastasis and therapy resistance have not only been identified in primary breast tumors, but they also occur at distal sights to regulate metastatic growth [26]. Based on this understanding of the ECM, there is mounting interest in developing therapies that target the stromal matrix [46-48].”
In this review,…..We will discuss the challenges that the TME poses to clinical therapies, specifically how the stroma drives immunosuppression and limits drug availability. Finally, we will highlight promising novel approaches to target the stroma in combination with immune modulation therapy or as direct molecular targets of ADC therapies.
On line 113. Are there differences in PD-L1 expression in the different molecular subtypes of triple-negative breast cancer? A table showing the expression profiling data of PDL1 in several types of tumors, it will be useful and informative.
---This is an excellent question. However, the only publication we identified on this topic did not define TNBC molecular subtypes using gene expression. It was based on IHC including AR IHC which does not correlate well with LAR subtype. Therefore, we did not revise to include this information. https://onlinelibrary.wiley.com/doi/epdf/10.1111/tbj.14110
(DOI: 10.1111/tbj.14110)
Line 230. In terms of matrix stiffness, it is important to mention how stiffness promotes the reorganization of tumor cell cytoskeleton and chromatin via cell-extracellular matrix and communication mechanisms such as SUN-KASH domain proteins to influence gene expression programs. This might explain, in part, the activation of immunosuppressive signaling molecules, in addition to the previously described variables (hypoxia, metabolic stress).
---We agree and thank the reviewer for this important comment. There are several in vitro studies demonstrating that matrix stiffness drives mechanical signaling (cytoskeletal reorganization that directly links to altered gene expression through the SUN-KASH complex) to activate immune suppression. This topic is very interesting and could be an entire review on its own. A deep dive into mechanical regulation in the tumor microenvironment is outside the scope of this review. However, we modified the text to briefly highlight stiffness signaling, especially the studies demonstrating that mechanical signaling leads to increased expression of PD-L1, immunosuppressive cytokines and matrix remodeling enzymes. Page 6: “Highly dense ECM increases tumor stiffness. The tumor cells respond to stiffness by activating mechano-signaling pathways that lead to increased expression of immune regulatory factors, such as PD-L1. CAFs also respond to the mechanical stiffness of the TME by activating feedforward mechanisms to further enhance ECM deposition, resulting in poor diffusion, increased hypoxia, and metabolic stress.”
Line 273. It will be very useful and informative if authors briefly describe the potential mechanisms of action of losartan in tumor cells or how the renin-angiotensin system is connected to the impacts of tumor perfusion, hypoxia, or ECM in the tumor microenvironment.
---We thank the reviewer for pointing out this important clarification. The text on page 8 has been modified to provide said clarification: “This reduction in ECM is due to the inhibition of TGFβ signaling, downstream of angiotensin II type 1 receptors, when losartan is bound.”
On line 349, where they mention that "anti-TGFβ treatment reduces the fibrous nature of the ECM". It is not clear what is indicated in Figure 1. In addition, it is important to mention which part of Figure 1 they are referring to.
---Figure 1 has been modified to improve the clarity and value of the information described.
In addition, the text on page 8 has been modified:
“Taking a slightly different approach, Takai, et al. observed that inhibition of TGFβ, a signaling molecule that that activates CAFs to secrete collagen leading to tumor fibrosis, may decrease tumor growth and metastasis.”
Table 1. Summary of Recent Trials for Novel Therapeutics in Metastatic TNBC. Please add the URL for the web page register in https://clinicaltrials.gov/, if any.
---We have revised summary Table 1 to include the URL for all trials. The hyperlinks have been added into caption for table.
Finally, feel that the review is very complex and involves several factors, so it would be more clear if authors divide the Figure 1 into two parts: i) addressing all the mechanisms of the tumour microenvironment and immune modulation, ii) the mechanisms of the tumour microenvironment and ADCs.
---Figure 1 has been modified to improve the clarity of therapeutic targets aimed at immune modulation (A-C) and ADCs targeting the ECM (D) and include mechanisms of action.
Reviewer 2 Report
The authors intended to describe in this review the impact of the tumor microenvironment on the effectiveness of therapies in ongoing clinical trials for triple negative breast cancer. However, the review mix between tumor stoma targets and different ongoing therapies in clinical trials in a confusing manner.
The review lacks a clear discussion about the different components in the tumor stroma and their effect on the therapies.
I suggest to focus on the different targets in tumor microenvironment, their influence in the respective treatments with the respective ongoing clinical trials.
There is no connection between the different items. For example, ADC treatment (4) and in microenvironment (5). Connect the 2 parts and avoid the repeated information.
I suggest to rewrite the manuscript in order to organize the important information you intended to provide.
Minor corrections:
Avoid using abbreviations if they appear once along the manuscript.
Figure 1 is very simplistic and lacks any added value. I suggest to create a more informative figure summarizing the different targets and treatments.
Add RAPA T cell immunotherapy.
Revise and correct misspelling.
Author Response
We thank the reviewers for their helpful comments, which have improved the clarity and overall flow of our manuscript. We have addressed the reviewers concerns in a point-by-point discussion below. The changes are also highlighted in red in the revised manuscript.
Reviewer #2:
The authors intended to describe in this review the impact of the tumor microenvironment on the effectiveness of therapies in ongoing clinical trials for triple negative breast cancer. However, the review mix between tumor stoma targets and different ongoing therapies in clinical trials in a confusing manner.
The review lacks a clear discussion about the different components in the tumor stroma and their effect on the therapies.
I suggest to focus on the different targets in tumor microenvironment, their influence in the respective treatments with the respective ongoing clinical trials.
There is no connection between the different items. For example, ADC treatment (4) and in microenvironment (5). Connect the 2 parts and avoid the repeated information.
I suggest to rewrite the manuscript in order to organize the important information you intended to provide.
We appreciate the reviewer’s comments on the structure. We considered several options for organization and integration of the clinical and basic science data. In review, we believe that our current structure provides the clearest review of the data. To address the reviewers concerns, we modified the introduction to clarify the scope of the review. We also focused on improving transitional statements and connecting topics to improve the flow of the manuscript. Changes to the manuscript are highlighted in red.
Minor corrections:
Avoid using abbreviations if they appear once along the manuscript.
Removed abbreviations for TNBC subtypes.
Figure 1 is very simplistic and lacks any added value. I suggest to create a more informative figure summarizing the different targets and treatments.
Figure 1 has been modified to improve the clarity of therapeutic targets aimed at immune modulation (A-C) and ADCs targeting the ECM (D) and include mechanisms of action.
Add RAPA T cell immunotherapy.
Thank you for bringing this to our attention. Although we considered including, we did not identify a way to fit this into the current flow of the manuscript and decided not to include this interesting, but preliminary approach.
ClinicalTrials.gov Identifier: NCT05144698
https://clinicaltrials.gov/ct2/show/NCT05144698
Revise and correct misspelling.
A thorough review of the manuscript was performed.
Round 2
Reviewer 2 Report
no suggestions